# CA-BIT: A Change Detection Method of Land Use in Natural Reserves

**Bin Jia [1,2], Zhiyou Cheng [1,3], Chuanjian Wang [1,3,*], Jinling Zhao [1,3]** and **Ning An [1,2]**

[1] National Engineering Research Center for Analysis and Application of Agro-Ecological Big Data, Anhui University, Hefei 230601, China
[2] School of Electronic Information Engineering, Anhui University, Hefei 230601, China
[3] School of Internet, Anhui University, Hefei 230039, China
* Correspondence: wcj_si@ahu.edu.cn

**Abstract:** Natural reserves play a leading role in safeguarding national ecological security. Remote sensing change detection (CD) technology can identify the dynamic changes of land use and warn of ecological risks in natural reserves in a timely manner, which can provide technical support for the management of natural reserves. We propose a CD method (CA-BIT) based on the improved bitemporal image transformer (BIT) model to realize the change detection of remote sensing data of Anhui Natural Reserves in 2018 and 2021. Resnet34-CA is constructed through the combination of Resnet34 and a coordinate attention mechanism to effectively extract high-level semantic features. The BIT module is also used to efficiently enhance the original semantic features. Compared with the overall accuracy of the existing deep learning-based CD methods, that of CA-BIT is 98.34% on the natural protected area CD datasets and 99.05% on LEVIR_CD. Our method can effectively satisfy the need of CD of different land categories such as construction land, farmland, and forest land.

**Keywords:** natural reserves; remote sensing; change detection; deep learning; residual attention network; bitemporal image transformer





## 1. Introduction

Natural reserves are the core carrier of ecological construction and occupy the primary position in safeguarding national ecological security. Anhui Province has implemented the Guidance on Establishing a Natural Reserves System with National Parks as the Mainstay [1] to build a natural reserves system with a reasonable layout, complete types, and perfect functions [2]. The rapid development of industrialization and urbanization has made the contradiction between ecological protection and economic development increasingly prominent. The fragmentation of environmental patches caused by human activities has greatly threatened biodiversity, which seriously affects the management, protection effect, and healthy development of natural reserves in Anhui Province. A large number of natural reserves with large area, wide distribution, complex geographical environment, complicated construction projects, and few supervision staff are present in Anhui Province. Timely detection and supervising of various illegal human activities in the natural reserves are difficult by traditional ground investigation means.

Remote sensing image change detection (CD) identifies changes in the Earth's surface by analyzing satellite images acquired at different times over the same geographical area [3]. Remote sensing image CD technology has always been widely utilized to record and monitor changes and maintain the sustainable development of the earth environment. At present, remote sensing image CD is widely used in many fields, such as urbanization detection [4], environmental monitoring [5–7], disaster assessment [8], and other fields.

The increasing popularity of high-resolution remote sensing images has expanded the potential applications of CD in high-resolution bitemporal images. At present, a deep convolutional neural network (DCNN) is successfully applied to high-resolution

remote sensing image analysis and CD tasks due to its strong advanced feature extraction capability [9]. The high-level semantic features of each temporal image are extracted on a CNN-based structure [10–12], and the final change map is generated by clustering or a threshold-based classification method. Obtaining contextual content over space and time is critical to identifying associated changes in multi-time high-resolution images. Thus, the latest CD models have been focused on increasing the receptive field, which is defined as the size of the area in the input where the feature is generated. Therefore, a CD model with stacked convolution layers [13,14] and the application of extended convolution and attention mechanism [15] is proposed. Currently, squeeze and excitation (SE) [16], the bottleneck attention module (BAM) [17], and convolutional block attention module (CBAM) [18] are mainly used on the mobile network design. However, SE only considers internal access information and ignores the importance of the spatial structure of the target in vision. BAM and CBAM attempt to introduce positional information through global pooling on the channels. However, this way can only capture local information rather than obtaining long-range dependent information. By contrast, the coordinate attention mechanism (CA) mechanism captures not only cross-channel information but also orientation perception and position-sensitive information [19]. The attention-based approach is effective in global information modeling but has difficulty relating remote spatiotemporal details.

The recent success of transformers (i.e., nonlocal self-attention) in natural language processing (NLP) has led researchers to apply transformers to various computer vision tasks. Chen Hao et al. [20] used the bitemporal image transformer (BIT) module, which can model the context information in token-based space–time, to enhance the original features. However, they failed to efficiently extract high-level semantic features. Bandara et al. [21] applied a layered transformer encoder (TE) with a lightweight *MLP* decoder to the CD task. The multi-level differential features can be effectively combined, but the spatiotemporal details cannot be efficiently linked.

In summary, we propose a CA-BIT model that combines the residual attention network (ResNet34-CA) and BIT for land use CD in natural reserves. Unlike BIT, our CA-BIT adds the CA to ResNet34 to improve image feature extraction and obtain better change recognition results. The model can provide automatic CD technical support for the daily supervision and environmental supervision of natural reserves. It is also important regarding the timely warning of ecological risks of natural reserves in Anhui Province.

The remainder of the paper is organized as follows. In Section 2, the architectural details of the proposed network are introduced. The study area, data, and experimental environment configuration are presented in Section 3. In Section 4, the proposed method is compared with other different deep learning models, and the CD results are analyzed. Some conclusions are presented in Section 5.

## 2. Network Model

### 2.1. CA-BIT Model Overview

The overall framework of the CA-BIT model is shown in Figure 1. First, the images $T_1$ and $T_2$ are input into the residual attention network (ResNet34-CA), and the feature map $X^i \in R^{H \times W \times C} (i \in 1, 2)$ is obtained for each image, where $H$, $W$, and $C$ denote the height, width, and channel size of the feature map, respectively. Next, the resulting feature map $X^i$ is fed into the BIT module to generate enhanced features $X_T^i$. Then, fusing the feature map $X^i$ yields the feature map $X_{new}^i$. The resulting feature map $X_{new}^i$ is fed to the prediction part to produce pixel-level predictions.

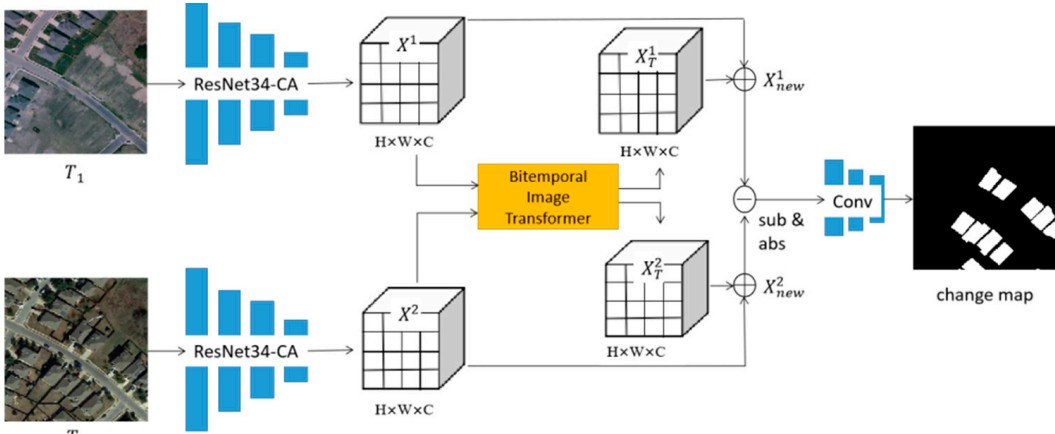

**Figure 1.** Framework of the CA-BIT model. We first use the image features extracted by the ResNet34-CA backbone. Then, we enhance the image features by the BIT module and fuse the features. Finally, our prediction part produces pixel-level predictions by feeding the computed the feature difference images into a shallow CNN.

### 2.2. Residual Attention Network

The backbone part of this network is to extract the bitemporal image feature maps through a ResNet34 network [22] combined with a CA residual module (Figure 2). We use CA to build the CA residual module, which adds CA after two layers of $3 \times 3$ convolution in the residual block structure. Then, we apply it to the ResNet34 network and construct it as ResNet34-CA to better extract the high-level semantic feature maps.

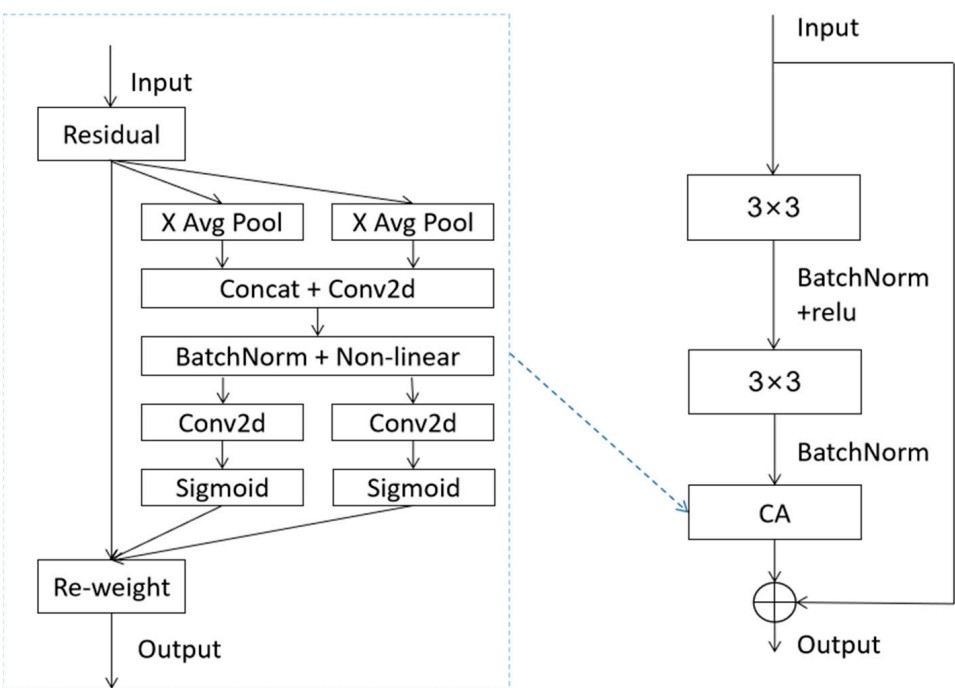

**Figure 2.** CA residual module. The CA is inserted into the residual block. Note: "X Avg Pool" and "Y Avg Pool" refer to 1D horizontal global pooling and 1D vertical global pooling, respectively.

### 2.3. Bitemporal Image Transformer

We refer to the BIT model [20], and feature fusion is added to the original BIT module. The overall block diagram is shown in Figure 3. The high-level concept of change objects of interest can be expressed in terms of several visual words, that is, semantic tokens. For this purpose, we represent bitemporal images as several tokens and use a transformer encoder

(TE) [23] to model the context in a compact token-based space–time. We then refine the original features through a transformer decoder (TD) and fuse the original features through a jump connection, where the learned context-rich markers are fed back into the pixel space.

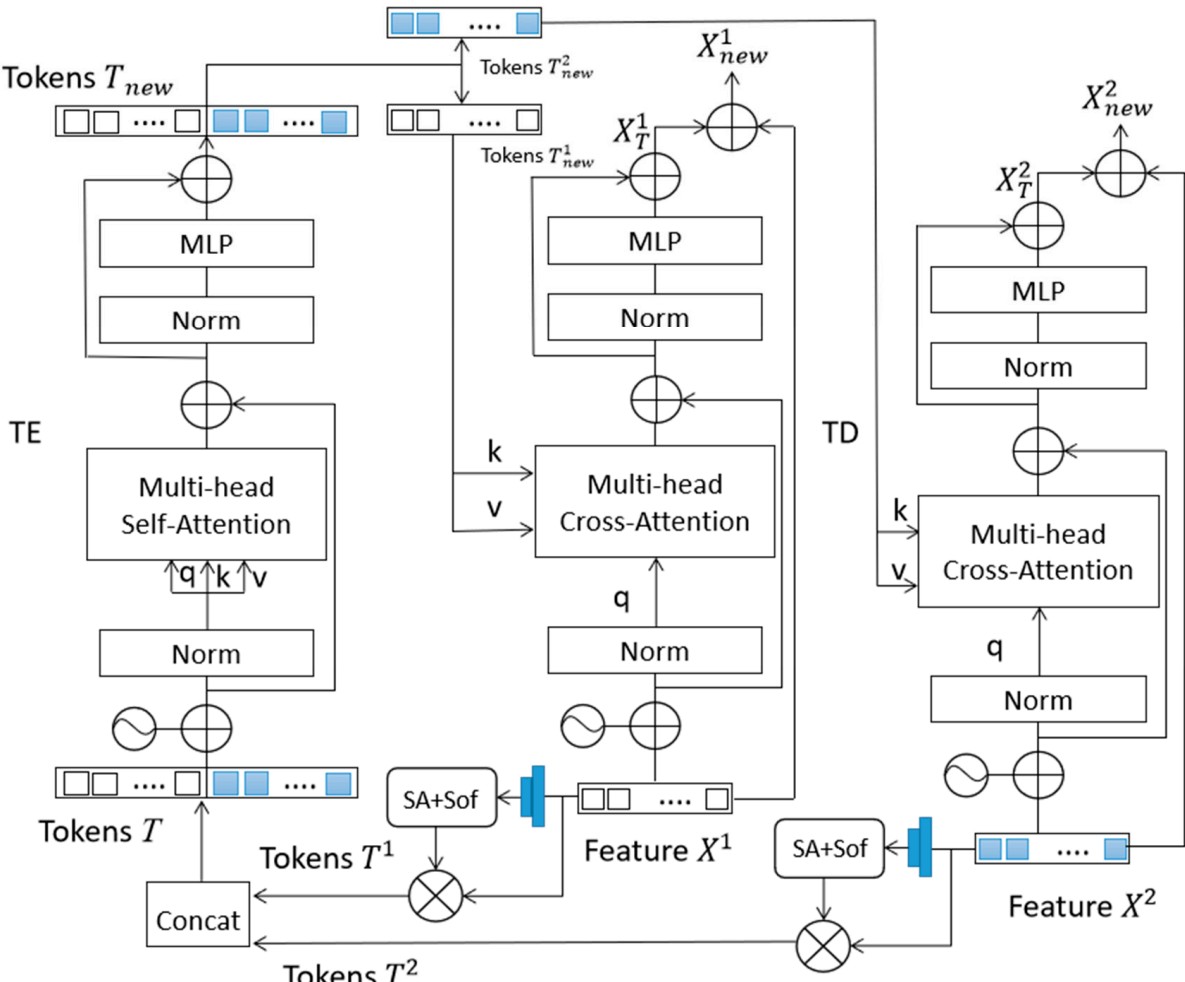

**Figure 3.** Bitemporal image transformer module and feature fusion.

The BIT learns a set of spatial attention (SA) maps by concentrating the feature mapping space into a set of features to obtain compact semantic tokens. Let $X^1, X^2 \in R^{HW \times C}$ be the bitemporal feature graph of the input. Let $T^1, T^2 \in R^{L \times C}$ be two groups of tokens, where $L$ ($L << HW$) is the size of the set of vocabularies for the token.

For each pixel on the feature map $X^1, X^2$, we utilize a point-wise convolution to obtain $L$ semantic groups, and each group represents one semantic concept. Then, the Softmax (Sof) function is used on the $HW$ dimension of each semantic group to calculate the SA map. The weighted average sum of the pixels in $X^1, X^2$ is calculated using the attention mapping to obtain a compact set of vocabularies of size $L$, that is, semantic tokens $T^1, T^2 \in R^{L \times C}$. The formula is as follows:

$$T^1 = (A^1)^T X^1 = (\sigma(\phi(X^1; W)))^T X^1 \tag{1}$$

$$T^2 = (A^2)^T X^2 = (\sigma(\phi(X^2; W)))^T X^2 \tag{2}$$

where $\phi(\cdot)$ indicates the point-wise convolution with the learnable kernel $W \in R^{C \times L}$, and $\sigma(\cdot)$ is the Sof function that normalizes each semantic group to obtain the attention map $A^1, A^2 \in R^{HW \times L}$. $T^1, T^2$ is calculated from the multiplication of $A^1, A^2$ and $X^1, X^2$.

### 2.3.1. Transformer Encoder

After concatenating the two semantic token sets $T^1, T^2$ into one token set $T \in R^{2L \times C}$, we then model the context between these tokens with the TE part. The TE part can utilize the global semantic relationships in token-based space–time to generate context-rich token representations for each time. As shown on the left side of Figure 3, the token $T$ is fed into the TE part to obtain a new token set $T_{new} \in R^{2L \times C}$.

The TE part consists of multi-head self-attention (MSA) and multi-layer perceptron (*MLP*) blocks. Different from the original transformer that uses the post-norm residual unit, we use the validated ViT [24] pre-norm residual unit, that is, the normalized layer is added before the MSA/*MLP*.

In each layer *l*, the self-attentive input is a triple (query *Q*, key *K*, and value *V*) that is calculated from the input $T^{(l-1)} \in R^{2L \times C}$ as follows:

$$Q = T^{(l-1)}W^q \tag{3}$$

$$K = T^{(l-1)}W^k \tag{4}$$

$$V = T^{(l-1)}W^v \tag{5}$$

where $W_q^{l-1}, W_k^{l-1}, W_v^{l-1} \in R^{C \times d}$ are the three learnable parameters of the linear projection layers, and *d* is the channel dimension of the three layers. An attention head is expressed as

$$Att(Q, K, V) = \sigma\left(\frac{QK^T}{\sqrt{d}}\right)V \tag{6}$$

where $\sigma(\cdot)$ represents the Sof function running on the channel dimension.

The MSA block of the TE part executes multiple stand-alone attention heads in parallel, and it connects the output and then projects it to obtain the final value. The formula is

$$\begin{aligned} MSA\left(T^{(l-1)}\right) \\ = Concat(head_1, \cdots, head_h)W^O \\ where\ head_j = Att\left(T^{(l-1)}W_j^q, T^{(l-1)}W_j^k, T^{(l-1)}W_j^v\right) \end{aligned} \tag{7}$$

where *head* is the number of attention heads, and $W_j^q, W_j^k, W_j^v \in R^{C \times d}, W^O \in R^{hd \times C}$ represent the linear projection matrix.

The *MLP* block of the TE part is composed of two linear transformation layers, with an activation of Gaussian error linear unit [25] in the middle. The dimension of input and output is *C*, and the inner layer is *2C*. The formula is

$$MLP\left(T^{(l-1)}\right) = GELU\left(T^{(l-1)}W_1\right)W_2 \tag{8}$$

where $W_1 \in R^{C \times 2C}, W_2 \in R^{2C \times C}$ represent the linear projection matrix.

### 2.3.2. Transformer Decoder

A new token set $T_{new} \in R^{2L \times C}$ is obtained from the TE part, split into two sets of context-rich tokens $T_{new}^1, T_{new}^2 \in R^{L \times C}$, and input into the TD part composed of multi-head cross-attention (MA) and *MLP*. The changes in interest are well revealed by the fact that these context-rich markers contain compact high-level semantic information. Then, we need to project the representation of the concept back into pixel space to obtain pixel-level features. Given a series of features $X^i (i = 1,2)$, the TD part utilizes the relationship between each pixel and the set of tokens $T_{new}^i$ to obtain the refined features $X_T^i$.

In *MA*, the keys and values are from the token set $T_{new}^i$, and the queries are from the image feature $X^i$. In each layer $l$, the *MA* is formally defined as

$$MA\left(X^{i,(l-1)}, T_{new}^i\right) = Concat(head_1, \cdots, head_h)W^O,$$
$$where\ head_j = Att\left(X^{i,(l-1)}W_j^q, T_{new}^i W_j^k, T_{new}^i W_j^v\right) \tag{9}$$

where $W_j^q, W_j^k, W_j^v \in R^{C \times d}, W^O \in R^{hd \times C}$ represent the linear projection matrix, and head is the number of attention heads.

The feature map $X^i$ ($i$ = 1,2) and refined features $X_T^i$ obtained from the TD part are fused to obtain the feature map $X_{new}^i$.

$$X_{new}^i = FeatureX^i + X_T^i \tag{10}$$

### 2.4. Prediction Part

The CA-BIT model extracts the resulting high-level semantic feature $X_{new}^i$, which uses a very shallow FCN for change recognition. The predictor head generates the predicted change probability map $P \in R^{H_0 \times W_0 \times 2}$ ($H_0$ and $W_0$ are the height and width of the original image, respectively) according to the feature map $X_{new}^1, X_{new}^2$, which is given by

$$P = \sigma(g(up(D))) = \sigma\left(g\left(up\left(\left|X_{new}^1 - X_{new}^2\right|\right)\right)\right) \tag{11}$$

where $D \in R^{H \times W \times C}$ is the element absolute subtraction value of two feature maps. Upsampling is conducted for $up : R^{H \times W \times C} \to R^{H_0 \times W_0 \times C}$, and the classifier is changed to $g : R^{H_0 \times W_0 \times C} \to R^{H_0 \times W_0 \times 2}$. $\sigma(\cdot)$ represents the Sof function.

### 2.5. Loss Function

In the training phase, the network parameters are optimized by minimizing the cross-entropy loss. The loss function is formally defined as

$$L = \frac{1}{H_0 \times W_0} \sum_{h=1,w=1}^{H,W} l(P_{hw}, Y_{hw}) \tag{12}$$

where $l(P_{hw}, y) = -\log(P_{hwy})$ is the cross-entropy loss, and $P_{hw}$ is the label of the pixel at the position (*h,w*).

## 3. Data and Experiments

### 3.1. Study Area and Data Sources

3.1.1. Overview of the Study Area

Anhui is a provincial administrative region of the People's Republic of China, which is located in the Yangtze River Delta region of East China (114°54′—119°37′, 29°41′—34°38′), with a total area of 140,100 km². The province has established more than 300 protected nature areas at various levels, including nature reserves, geological parks, scenic spots, forest parks, and wetland parks. These areas play an important role in the conservation of biodiversity and protection of natural heritage, which improve ecological environment quality and maintain ecological security.

3.1.2. Experimental Data

The experimental data come from the satellite remote sensing detection project of human activities in natural reserves in Anhui Province, including the high-resolution remote sensing images and the human activity vector database of natural reserves in Anhui Province. The data comprise the high-resolution remote sensing image of Beijing No. 2 acquired in 2018 with a spatial resolution of 1 m and the high-resolution remote sensing image of Gaojing No. 1 acquired in 2021 with a spatial resolution of 0.5 m. Owing

to the huge amount of data, the study sample area is reasonably selected according to the distribution of the change type, and the data contain 685,703,168 pixels. The map of Wanfoshan–Longhekou Reservoir (Wanfo Lake) in Shucheng County, Lu'an City, Anhui Province is synthesized using high-resolution data and made by ArcGIS10.4 software. It is shown in Figure 4, and the red box is the sample area (Figure 5).

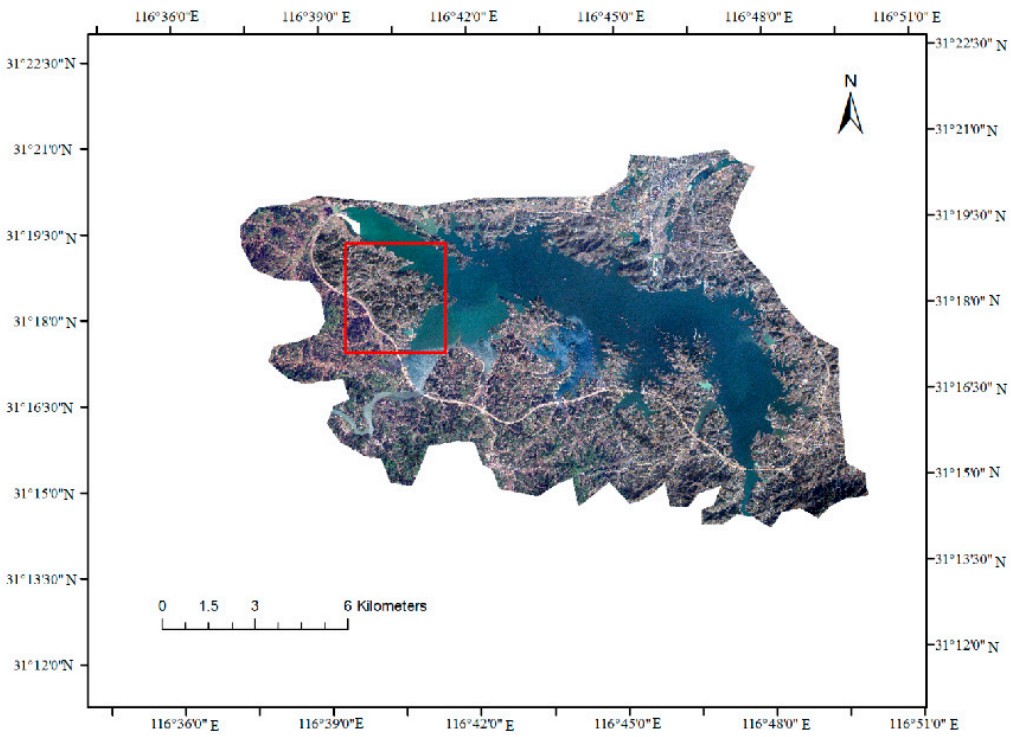

**Figure 4.** Wanfoshan–Longhekou Reservoir (Wanfo Lake) Scenic Area in Shucheng County, Lu'an City, Anhui Province (Beijing No. 2).

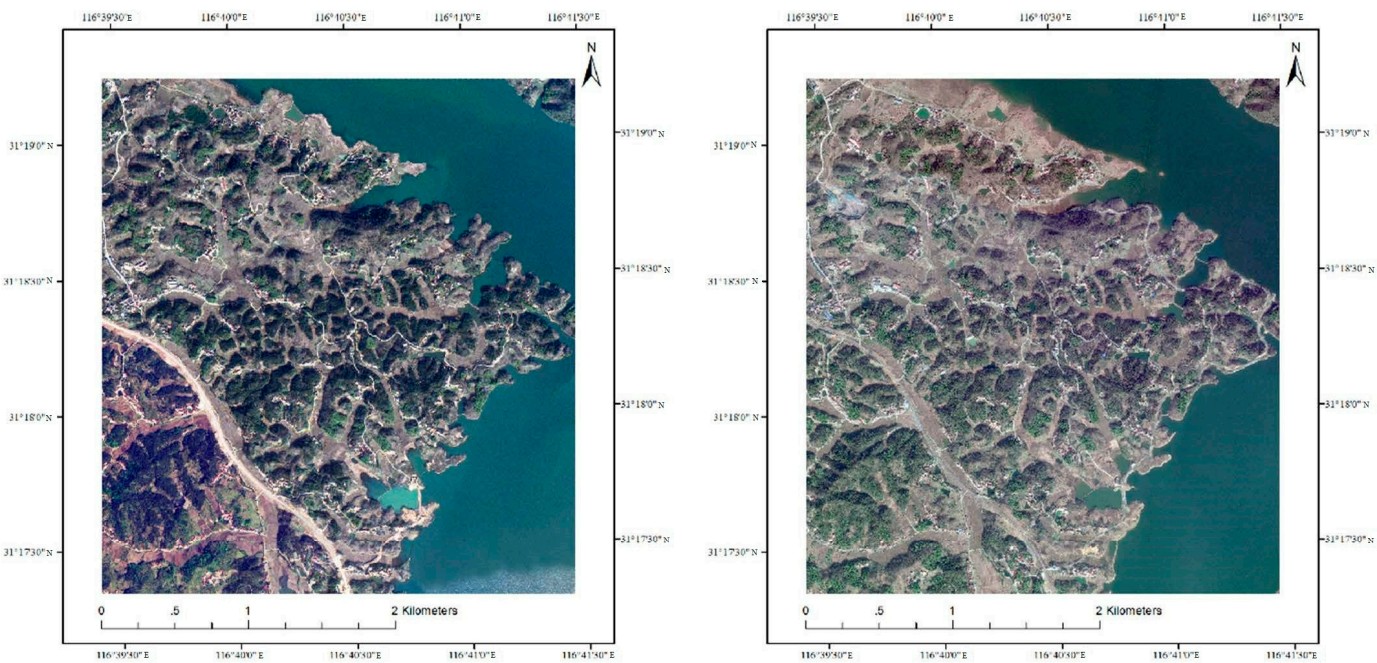

**Figure 5.** Data of sample area.

The reference basis of land use types in natural reserves in Anhui Province is the Technical Specification for Remote Sensing Monitoring of Human Activities in natural reserves (HJ1156-2021). Considering the actual needs of the Anhui natural satellite remote sensing detection project, 10 different classes are distinguished (mineral resources development, industrial development, energy development, tourism development, transport development, aquaculture development, agricultural development, settlements and other activities, forest management, and engineering construction). Then, by adjusting the change map spot of natural reserves, two types of label data for the change/no change in natural reserves are made, in which the numbers of changed and unchanged pixels are 26,316,288 and 659,386,879, respectively. The remote sensing images and label data are trimmed to a size of 256 pixels × 256 pixels with no overlap, and they are randomly divided into the training set, test set, and validation set in a ratio of 7:2:1, with 7322, 2092, and 1046 pairs of data blocks, respectively.

We also conduct experiments on another public CD dataset. The LEarning, VIsion and Remote sensing CD (LEVIR_CD) is a large-scale public building CD dataset. It contains 637 ultra-high-resolution (VHR, 0.5 m/pixel) Google Earth image patches on a size of 1024 pixels × 1024 pixels. Owing to GPU memory capacity limitations, we cut the image into small blocks of a size of 256 pixels × 256 pixels with no overlap and randomly divide the dataset (train, test, and validation). Therefore, we obtain 7120, 2048, and 1024 pairs of blocks for train, test, and validation, respectively.

### 3.2. Experimental Environment Configuration and Evaluation Indicators

#### 3.2.1. Experimental Environment Configuration

Our proposed model uses PyTorch as a deep learning framework and uses JetBrains PyCharm 2020 as the development platform; the development language is Python3.8, and all models are trained and tested on computers configured as Intel Core (TM) i7-10700K CPU and NVIDIA GeForce GTX 3080 Ti graphics cards. The same experimental parameters are utilized in the experiments, including momentum (0.99), weight decay (0.0005), batch size (8), running epochs (200), and initial learning (0.01). The gradient descent optimization method used for the optimization model is stochastic gradient descent [26]. The data enhancement strategy includes random flipping, rotating, and Gaussian blur while loading the image pairs. For simplicity, the best model validated after each training phase is used to evaluate the test set.

#### 3.2.2. Evaluating Indicator

We use the accuracy of the evaluation index precision, recall, harmonized mean of precision and recall (*F1*), intersection ratio (*IoU*), and overall accuracy (*OA*) to evaluate the accuracy for quantifying the effect of the evaluation model. The calculation formula of the evaluation index is

$$precision = \frac{TP}{TP + FP} \tag{13}$$

$$recall = \frac{TP}{TP + FN} \tag{14}$$

$$F1 = \frac{2}{recall^{-1} + precision^{-1}} \tag{15}$$

$$IoU = \frac{TP}{TP + FN + FP} \tag{16}$$

$$OA = \frac{TP + TN}{TP + TN + FN + FP} \tag{17}$$

where *TP*, *TN*, *FP*, and *FN* express the number of true positive, true negative, false-positive, and false-negative, respectively.

## 4. Experimental Results and Analysis

Six models, namely, full convolutional Siamese difference (FC-Siam-Di), full convolutional Siamese connection (FC-Siam-Conc), dual-task constrained deep Siamese convolution network (DTCDSCN), ResNet34_CA network, BIT, and transformer-based Siam network (ChangeFormer), are selected for comparison to verify the confidence of our CD model. We use our natural reserve CD dataset and the LEVIR_CD to verify the above-mentioned CD network.

The results of the accuracy evaluation of the various methods are shown in Table 1. The sample area is analyzed qualitatively to show the results of the CD task more intuitively. Figure 6 illustrates the CD results of the sample areas of the natural reserves. Figure 6a shows the results of visual interpretation of reference changes based on high-resolution remote sensing imagery. Figure 7 illustrates the CD results of the public building CD dataset LEVIR_CD.

**Table 1.** Evaluation of CD accuracy under different methods.

| Method | Natural Reserve CD | | | | | LEVIR_CD | | | | |
|---|---|---|---|---|---|---|---|---|---|---|
| | Precision | Recall | F1 | IoU | OA | Precision | Recall | F1 | IoU | OA |
| FC-Siam-Conc | 37.25 | **64.76** | 47.3 | 30.98 | 96.02 | 91.99 | 76.77 | 83.69 | 71.96 | 98.49 |
| FC-Siam-Di | 39.18 | 50.8 | 44.24 | 28.41 | 96.47 | 89.53 | 83.31 | 86.31 | 75.92 | 98.67 |
| ResNet34-CA | 43.55 | 44.68 | 44.11 | 28.29 | 96.87 | 86.13 | 80.63 | 83.29 | 71.36 | 98.35 |
| BIT | **69.37** | 41.55 | 51.97 | 35.11 | 97.20 | 89.24 | **89.37** | 89.31 | 80.68 | 98.92 |
| DTCDSCN | 52.30 | **73.3** | **61.04** | **43.93** | 97.42 | 88.53 | 86.83 | 87.67 | 78.05 | 98.77 |
| ChangeFormer | 68.55 | 53.89 | 60.34 | 43.21 | **98.04** | **92.05** | **88.80** | **90.40** | **82.48** | **99.04** |
| CA-BIT | **74.61** | 60.32 | **66.71** | **50.05** | **98.34** | **92.30** | 88.72 | **90.48** | **82.61** | **99.05** |

Note: All values are reported as a percentage (%). Black in bold indicates the best, and blue in bold is the second.

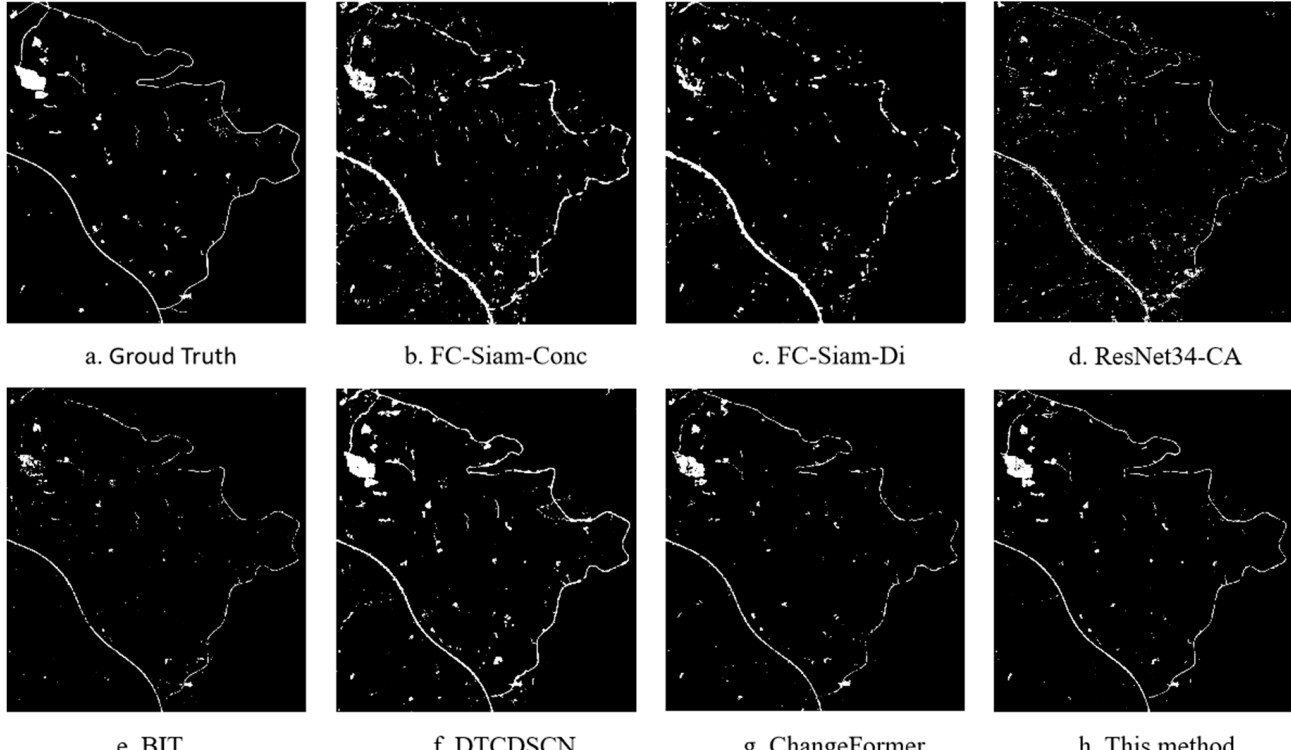

a. Groud Truth   b. FC-Siam-Conc   c. FC-Siam-Di   d. ResNet34-CA

e. BIT   f. DTCDSCN   g. ChangeFormer   h. This method

**Figure 6.** CD results under different methods (sample area).

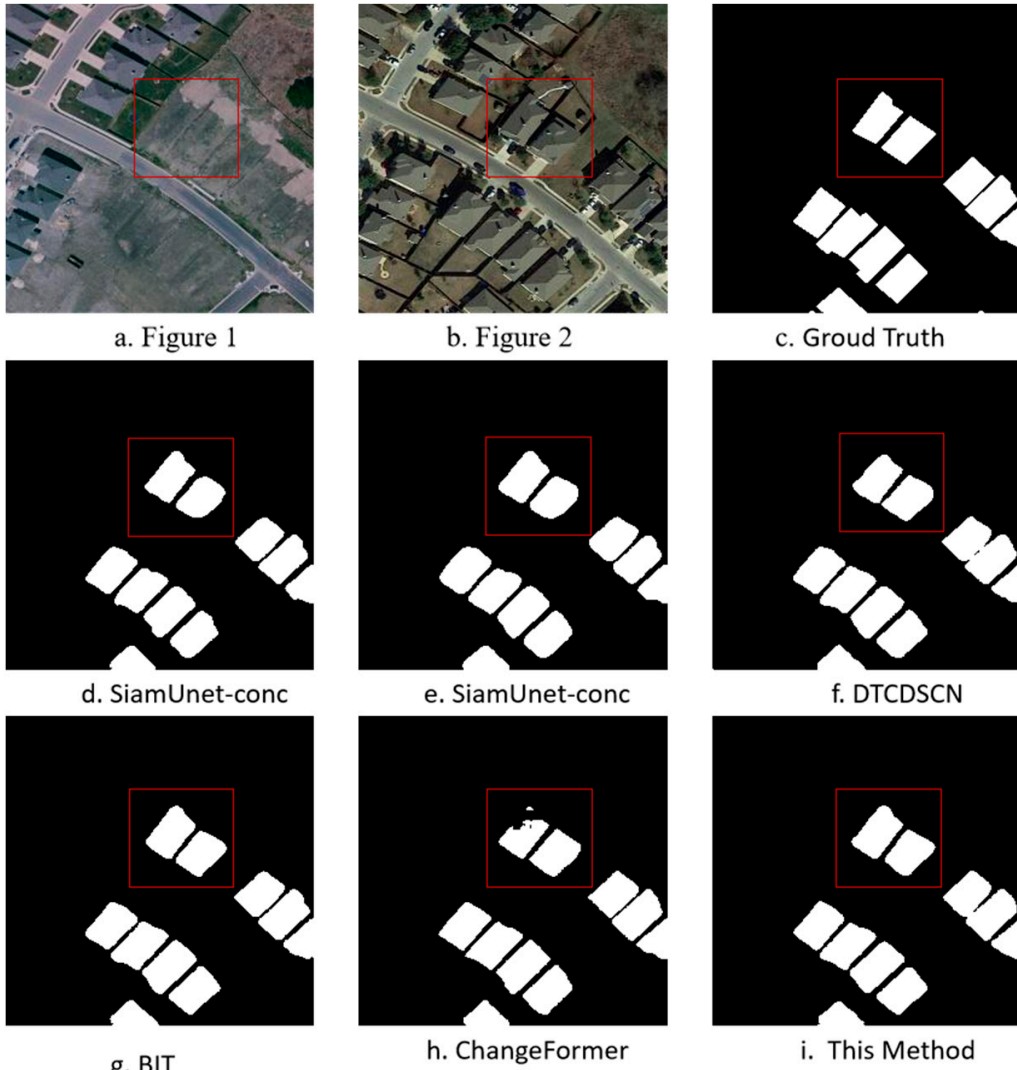

**Figure 7.** CD results under different methods (LEVIR_CD).

As observed from the comparison effect of Figure 6 and Table 1 in the natural reserves CD dataset, the early full convolution of Siamese (difference and connection) network changes has the lowest detection accuracy. The results of the change identification can only roughly extract the approximate extent of the change, but it is difficult to obtain detailed results of the change of real geographical entities. The ResNet34-CA network with attention mechanism added to the residual network increases the receiving domain but fails to effectively connect the dual time characteristics, which results in low accuracy, and large blocks of change areas are under-judged in change recognition. The BIT network adds a multi-layer transformer after the CNN, and it models the context information in token-based space–time to enhance the original semantic features. However, it ignores the effective extraction of high-level semantic features of the image. Obtaining good boundaries is difficult due to the dramatically changing areas (Figure 6e); the $F$1 is 0.5197 and the *IoU* is 0.3511. DTCDSCN uses a multi-scale feature connectivity approach to increase channel attention and SA in the deep Siamese FCN, which results in more distinguished features and higher CD accuracy. There is a clear edge profile and also some misjudgements in the change recognition. The transformer-based Siamese network ChangeFormer uses a hierarchical TE and a simple *MLP* decoder to effectively utilize the dual-temporal image multi-level differential features to achieve high overall accuracy, delicately extract areas of change, and obtain good boundaries. The CD accuracy of the CA-BIT algorithm is the best

overall with an *OA* of 0.9834, an *F*1 of 0.6671, and an *IoU* of 0.5005, and obtains detailed, realistic results of changes in geographical entities.

Table 1 shows that the present model still achieves better CD accuracy in the LEVIR_CD public datasets than the latest ChangeFormer networks. As shown in Figure 7, compared with the detailed performance in the red box, the present model can better express the small ground material changes, which further proves its effectiveness and feasibility.

In conclusion, the CA-BIT method can be used in nature reserve CD datasets. This model can effectively exert the advantages of high-resolution data to enrich spatial information. It can also obtain good CD results by extracting the semantic features and enhancing bitemporal features. The proposed model has achieved a relatively good CD effect compared with other methods. However, it also needs to be improved in the actual application of human activity detection in natural reserves.

## 5. Conclusions

Natural reserves have many change categories, and the number of change samples is much smaller than unchanged samples. To address these problems, we propose a remote sensing image CD model CA-BIT that combines ResNet34-CA and BIT. The CA-BIT model not only effectively extracts the global semantic features but also models the context information in token-based space–time to enhance the original semantic features. As a result, it reveals the changes in interest in the presence of dual-temporal images. In the natural conservation area CD datasets, the CA-BIT model works better than other recent deep learning-based models. The CA-BIT model still has better applicability and robustness in the public CD dataset LEVIR_CD.

**Author Contributions:** Conceptualization, B.J., Z.C. and C.W.; methodology, B.J., Z.C. and C.W.; formal analysis, C.W. and J.Z.; data curation, C.W.; writing—original draft preparation, B.J.; writing—review and editing, Z.C., J.Z. and N.A.; visualization, N.A.; supervision, Z.C. and C.W.; funding acquisition, C.W. and N.A. All authors have read and agreed to the published version of the manuscript.

**Funding:** This research was funded by the Natural Science Foundation of China (31971789), the Excellent Scientific Research and Innovation Team (2022AH010005), and the National Key Research and Development Project (2017YFB050420).

**Institutional Review Board Statement:** Not applicable.

**Informed Consent Statement:** Not applicable.

**Data Availability Statement:** https://justchenhao.github.io/LEVIR/ (accessed on 10 January 2023).

**Acknowledgments:** We thank all editors and reviewers for their valuable comments and suggestions, which improved this manuscript.

**Conflicts of Interest:** The authors declare no conflict of interest.

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
