# Peer review of "CA-BIT: A Change Detection Method of Land Use in Natural Reserves"

_agronomy, doi:10.3390/agronomy13030635_

Round 1

Reviewer 1 Report

Generally, the structure of the paper is appropriate. The article's subject matter is essential and relevant. Authors should take more care of the editing side of the article, e.g., missing author (line 4), "Transformer" written with a capital letter, introducing own name of Softmax function (sof). In my opinion description of the proposed network model is hard to follow and could be written in a more clear way.

Reviewer 2 Report

OBSERVATIONS TO MANUSCRIPT: CA-BIT: A change detection method of land use in natural reserves

The document is interesting, since it establishes a methodological scheme, where remote sensing technology is used to identify dynamic changes in land use, in order to timely warn of ecological risks to natural resources, which could provide technical support for its management.

However, I have some observations to improve the manuscript.

You must follow the indications that the journal provides in the instructions for authors.

Please throughout the document, you write the references correctly. References should be numbered in order of appearance and indicated by a numeral or numerals in square brackets, for example [1] or [2,3] or [4-6], according to the instructions for authors.

Introduction.

In the final part of the introduction, you mention the parts of your document, without mentioning assumptions or objectives of your work. Please, authors must follow the instructions for authors.

On the other hand, the authors do not explain the foundations that they considered to propose the union of ResNet34-CA and the BIT module, to use it in the extraction of semantic characteristics and its subsequent modeling to improve the original semantic characteristics, since you only cite some works where models were used separately. This is important, because you would establish the assumptions or hypotheses of your research and therefore its objectives.

Part of section two are fundamentals of the models that the authors intend to unite, which could be part of the introduction; while another part, the authors explains the changes that they would make to the models, which would correspond to materials and methods.

Section three corresponds to materials and methods.

You must adequately write the methods you used to justify the results you obtained.

In the section experimental results and analysis, you repeat part of what you mentioned in the previous sections. On the other hand, you indicate the results; however, you do not explain or discuss them, you must discuss the results.

Reviewer 3 Report

The manuscript suggests an algorithm to detect changes on the ground using bi-temporal satellite images.

The manuscript overall is understandable. However, there are some critical issues that I suggest to addresses. 

The data preparation part is not clear to me. For instance, how many images are there eventually and how they are generated. 

The training part is slightly touched. This step must be explained in more detail. For instance, the number of epochs, training time, the criteria for selecting the best weights, etc. must be explained.

The authors use acronyms too many times. I suggest avoiding them as much as they can. Even for CD, if they use “change detection” the manuscript will be more readable. 

Overall, the authors used just two datasets to compare their model with other deep learning models and concluded that their model works better. I do not think we can conclude such a claim using just two datasets. I suggest increasing the number of datasets or providing solid references that used just a few datasets to approve that their model working better than others. 

There are more comments in the pdf file.

Reviewer 4 Report

Report Reviewer

Summary brief

The manuscript examined a method for identifying land use in nature reserves using high-resolution remote sensing images. The target objects that differ from the soil are identified using the change detection method based on the improved BIT model (CA-BIT). The work exploits some deep learning algorithms and evaluates the level of accuracy of the predictive models. Overall, the work focuses on an emerging research topic and exploits some very promising image analysis techniques. However, while it is not strictly based on agronomic topics of interest, it is a topic that allows for the identification of land use.

General concept comments

The article is comprehensive and a very good and understandable summary of the machine learning methods and especially the model used has been made. The writing in English was well applied and the use of technical words. The experimental scheme was well taken care of. However, the introductory part needs to be supplemented with scientific developments in the field. Several scientific remarks were made to improve the manuscript.

In the introduction, rows 52 to 56: Deep Convolutional Neural Networks, DCNN and Residual Attention Network (ResNet34-CA) must be implemented. Expanding the description of the receiving domain (RF).

The description of the materials and methods is well-detailed and allows a full understanding of the experimental conditions, experimental schemes and techniques used. Its can be reply in other environmental conditions in according to the present literature. 

-Row 148: Maybe you mean Soft function?

-Row 172: Is it appropriate to also evaluate a fuzzy probability map, to compare with your method?

-Row 173: Why was FCN applied as a segmentation method, replacing fully connected layers with multiple convolutional layers?

-Rows 198-199: Specify the source from which the images were downloaded. In addition to the resolution, provide the type of format, whether single-band or multi-band. Furthermore, it is not clear why you chose two different resolutions (1m and 0.5m) for the two years of analysis.

The results are not fully comprehensive, in fact, it is recommended to supplement the comments on Table 1 for better understanding. The discussion of the data is insufficient and requires corrections and supplements.

Row 277: Did this result affect the process of extracting global semantic features from images?

Row 284: Check if other authors who have used similar models have obtained concordant results.

Rows 293-294: Could it be that natural reserves are influenced not only by the number of samples, but nevertheless by characteristics of the target shape? More on image analysis, insert bibliographical references.

Round 2

Reviewer 2 Report

OBSERVATIONS TO MANUSCRIPT: CA-BIT: A change detection method of land use in natural reserves

The document has improved a lot. The approach is clear and the discussion is more precise; however, you must attend to the following observation.

In writing the document, you should make some precisions. When you write the references of the sentences you quote, you write the reference after a comma or a point and even next to the final word of the sentence. For example, in the sentence: Remote sensing image change detection (CD) identifies changes in the Earth's surface by analyzing satellite images acquired at different times over the same geographical area.[3] Remote sensing image CD technology has always been widely used to record and monitor the change and maintain the sustainable development of the earth environment. At present, remote sensing image CD is widely used in many fields, such as urbanization detection,[4] environmental monitoring,[5-7] disaster assessment,[8] and other fields.

References should be written as follows: Remote sensing image change detection (CD) identifies changes in the Earth's surface by analyzing satellite images acquired at different times over the same geographic area [3]. Remote sensing image CD technology has always been widely used to record and monitor the change and maintain the sustainable development of the earth environment. At present, remote sensing image CD is widely used in many fields, such as urbanization detection [4], environmental monitoring [5-7], disaster assessment [8] and other fields.

Between the last word of the sentence and the reference there must be a space followed by the punctuation (comma or point), as the case may be; If there is no comma or point, you should only leave the space between the word and the reference. Same situation when writing the last name of the first author and collaborators, after the point of et al. you should only leave the space.

Please, you must show consistency throughout the document.

Author Response

 Thank you for this valuable feedback. We followed the example you gave us to carefully fix the editing errors we made and to avoid the same problems.

Reviewer 3 Report

The authors addressed some of the comments. However, there are lots of minor comments that I mentioned in the previous review and still are not addressed. I would like them to change it or let me know the reason why they do not like to change it. It will help both them and me in our future research to perform better either in writing or reviewing jobs. 

Author Response

We are very sorry that we did not notice some comments (marked fields) when we were reviewing your review comments. Thank you for your valuable comments, we will make careful corrections and show the final result in the revised manuscript.

Reviewer 4 Report

REPORT REVIEW _ second round

The manuscript has been greatly improved, but still needs minor editing. The introduction is well done but a little concise, so it needs to be improved.  Are the results obtained just a correlation matrix? Can this section and the discussion be improved at the same time?

Other small suggestions are:

- Line 47-48: The acronym is wrong “deep convolutional neural network (CNN)”. Probably it is DCNN.

Figure 4. The thickness of the red line of the box is little.

Figure 5. The coordinates are missing and the scale bar must correspond to the model in figure 4.
